

# Technical note: Large offsets between different datasets of seawater isotopic composition:

## an illustration of the need to reinforce intercalibration efforts

Gilles Reverdin[1], Claire Waelbroeck[1], Antje H. L. Voelker[2,3], Hanno Meyer[4]

[1] Laboratoire LOCEAN/IPSL, Sorbonne Université-CNRS-IRD-MNHN, Paris, 75005, France

[2] Instituto Português do Mar e da Atmosfera (IPMA), Algés, Portugal

[3] Center of Marine Sciences (CCMAR), Universidade do Algarve, Faro, Portugal

[4] Alfred Wegener Institute Potsdam, 14473 Potsdam, Germany

Corresponding author: Gilles Reverdin gilles.reverdin@locean.ipsl.fr

Abstract
We illustrate offsets in surface seawater isotopic composition between recent, public
datasets from the Atlantic Ocean and the subtropical southeastern Indian Ocean. The
observed offsets between datasets often exceed 0.10‰ in $\delta^{18}O$ and 0.50‰ in $\delta^2H$. They
might in part originate from different sampling of seasonal, interannual or spatial
variability. However, they likely mostly originate from different instrumentations and
protocols used to measure the water samples. Estimation of the systematic offsets is
required before merging the different datasets in order to investigate spatio-temporal
variability of isotopic composition in the world ocean surface waters. This highlights the
need to actively share seawater isotopic composition samples dedicated to specific
intercomparison of data produced in the different laboratories and to promote best
practices, a task to be addressed by the new SCOR working group 171.


## 1. Introduction

Seawater isotopic composition ($^{18}O/^{16}O$ and $^2H/^1H$ ratios expressed as $\delta^{18}O$ and $\delta^2H$ in
‰ in the VSMOW/SLAP scale) is classified as an Essential Ocean/Climate Variable
(EOV/ECV) in international programs such as GEOTRACES and GO-SHIP. Stable
seawater isotopes ($\delta^{18}O$, $\delta^2H$) are used to trace sources of freshwater (precipitation,
evaporation, runoff, melting glaciers, sea ice formation and melting), both at the ocean
surface and in the ocean interior (Schmidt et al., 2007; Hilaire-Marcel et al., 2021).
Except for fractionation during phase changes, the water isotopic composition is nearly
conservative in the ocean.
A major emphasis is on high latitude oceanography. There, continental (or iceberg)
glacial melt, formation or melt of sea ice, and high-latitude river inputs (for the Arctic)
leave imprints on the surface ocean isotopic composition, as well as below the surface
down to 800 m close to ice shelves in the Southern Ocean (Randall-Goodwin et al., 2015;
Biddle et al., 2019, Hennig et al., 2024). In contrast, few studies have been performed on
the isotopic signature in the deep ocean (e.g., Prasanna et al., 2015; Voelker et al., 2015).
Seawater isotopes in the upper ocean at low latitudes are often vital for paleoclimatic
studies, as they are needed to calibrate proxies of past ocean variability in marine
carbonate records such as corals and foraminifera (e.g., PAGES CoralHydro2k working
group; Konecky et al., 2020). Seawater isotopes are also important tracers in the coastal
ocean, with emphasis on upwelling (Conroy et al., 2014, 2017; Kubota et al., 2022; Lao et
al., 2022), and river discharges (e.g., Amazon) (Karr and Showers, 2001). Surface ocean
seawater isotopes are also used to characterize evaporation rates and air-sea
interactions (Benetti et al., 2017).
The isotopic signatures of these different processes are evolving in our warming world,
which will imprint on the seawater isotopic composition (Oppo et al., 2007).
Additionally, seawater isotope data provide model boundary conditions and allow the
assessment of model performance in isotope-enabled Earth system models (e.g. Schmidt
et al., 2007; Brady et al., 2019; Cauquoin et al., 2019), thereby improving climate model
projections of the future.
Stable seawater isotope data have thus been massively produced in the last decades by
a variety of methods. For example, most data compiled in the "GISS Global Seawater
Oxygen-18 Database -V1.21" for stable seawater isotopes (LeGrande and Schmidt, 2006)
originate from Isotope-ratio Mass Spectrometry (IRMS). They were mostly measured in
earlier decades by dual-inlet technology (highest precision), whereas, more recently, the
continuous-flow method (lower precision) became widespread for seawater isotope
analysis. In the last decade, cavity ring-down spectroscopy (CRDS) turned into another
commonly used method as it allows parallel measurement of $\delta^{18}O$ and $\delta^2H$, but with
often lower precision, at least early on (e.g., Voelker et al., 2015).
Reverdin et al. (2022) recently compiled a mix of data produced by IRMS and CRDS at
LOCEAN (https://www.seanoe.org/data/00600/71186/). As CRDS and other laser
techniques (Glaubke et al., 2024; hereafter GWS2024) have become more prevalent
recently, they contribute a significant part of the new data produced and thus also to the
soon to be released CoralHydro2k seawater database for $\delta^{18}O$ ($\delta^2H$) with a focus on the
tropics (35°N-35°S) (Atwood et al., 2024).
There are potential differences between the data produced by the two methods.
Typically, $CO_2$-water or $H_2$-water equilibration was used for the IRMS measurements
and yields measurements of the activity of water, which decreases with increasing
salinity. Furthermore, concentration of divalent cations like $Mg^{++}$ are responsible
for slight changes in fractionation factors. On the other hand, the laser methods such as
CRDS evaporate the entire sample. If the samples have not been distilled beforehand,
there is an issue of salt deposition and of resulting absorption or desorption of water
with fractionation effects. In the LOCEAN database (Reverdin et al., 2022), an attempt
was made to adjust the data, based on the analysis of Benetti et al (2017b). This was also
adopted by at least one other group (Haumann et al., 2022), but overall, there is the
possibility of an offset of these data with respect to the ones of other groups using CRDS.
However, it should be noted that some studies reporting unadjusted $\delta^{18}O$ measurements
from CRDS and IRMS technique with $CO_2$-water equilibration provide data that were
undistinguishable within instrumental precision (Walker et al., 2016; Hennig et al.,

92    2024).

It is actually quite common when using water isotope data in studies involving more
than one dataset, to first evaluate whether there are possible offsets. Intercomparison
with earlier data or reference materials was a prerequisite for GEOTRACES sampling
campaigns, although for the water isotopes this was, unfortunately, seldomly followed
(e.g., Voelker et al., 2015). These intercomparisons often outline systematic differences
which could result from the issue outlined above, or from other issues, such as
uncertainties in reference materials used, analysis protocols, or isotopic changes in the
samples during their handling and storage (Benetti et al., 2017a; Akhoudas et al., 2019;
Hennig et al., 2024). In other cases, this was not done, either because the data stood by
themselves (Bonne et al., 2019, for $\delta^{18}O$ and $\delta^2H$ data), or there was no comparison data
available in the same region (GWS2024, for $\delta^{18}O$ data). The possible offsets can however
become an issue, when these data are placed in a larger context. For example, GWS2024
identify a large difference in the S-$\delta^{18}O$ relationship in the subtropical Indian Ocean
between their data in the southeastern part and other data in the southwestern Indian
Ocean. They also discuss and question differences in the deep water-masses isotopic
values between separate datasets, but as these might also be explained by large
uncertainties in these data, we will not address them further.
Using these two examples (Bonne et al., 2019; GWS 2024), the aim of this note is to point
out the interest when producing a new dataset, of exchanging collected samples to carry
a direct comparison, or, if this was not done, to compare the data with other published
data and evaluate potential systematic differences.
2. Comparisons
For identifying possible offsets, we consider surface ocean subsets of the LOCEAN data
base in specific regions for roughly the same years as the other data collected. The data
extracted are from the same regions as in the datasets of the two studies and are
gathered in S-$\delta^{18}O$ space as well as in S-$\delta^2H$ space (only presented for the Bonne et al
(2019) dataset), where S is reported as a practical salinity with the practical salinity
scale of 1978. The assumption done here as in many papers is that the S-$\delta^{18}O$
relationship holds on fairly large scales in the surface layer (for the eastern subtropical
North Atlantic, see for example, the discussion in Voelker et al (2015) and in Benetti et
al. (2017a)). Obviously, this has limitations, such as in areas influenced by more than
one water mass or by multiple freshwater end-members (meteoric, continental run-off,
sea ice melt or formation, evaporation).
2.1 Daily surface data collected from R.V. Polarstern
The surface seawater samples originated from daily collection during two years on
board RV Polarstern in 2015-2017 (Bonne et al., 2019). There is no salinity provided
with the data, and here we chose to associate them with the simultaneously collected
thermosalinograph (TSG) data collected on board the RV Polarstern and available from
PANGAEA (for each cruise, an indexed file with title starting by 'Continuous
thermosalinograh oceanography along Polarstern' is included in PANGAEA: for example,
TSG data for the first cruise (PS90) associated with the isotopic seawater data are found
at https://doi.org/10.1594/PANGAEA.858885). The water samples were not collected
from the same water line and pumping depth as the TSG data, which can result in
differences. This is however likely to be small in most circumstances away from large
freshwater input at the sea surface, such as from melting sea ice, intense rainfall and
river estuaries (Boutin et al., 2016).  We also applied an adjustment of +0.25‰ to the
$\delta^{18}O$ data of Bonne et al. (2019), based on post-analysis identification of a bias in an
internal reference material.
We then estimate averages of all the data as a function of salinity in two domains
extending poleward of the subtropical salinity maximum toward the higher latitudes in
the eastern part of the Atlantic Ocean (thus, 20°N to 65°N and the same in the southern
hemisphere). This is done by sorting out the data by salinity classes of 0.5. The LOCEAN
data until 2016 in the North and tropical Atlantic were presented by Benetti et al
(2017a), showing the tightness of the S-$\delta^{18}O$ and S-$\delta^{2}H$ relationships in vast domains of
the eastern Atlantic. In the North Atlantic, LOCEAN data have been continuously
collected since 2011, and south of 10°S in the eastern Atlantic mostly since 2017.

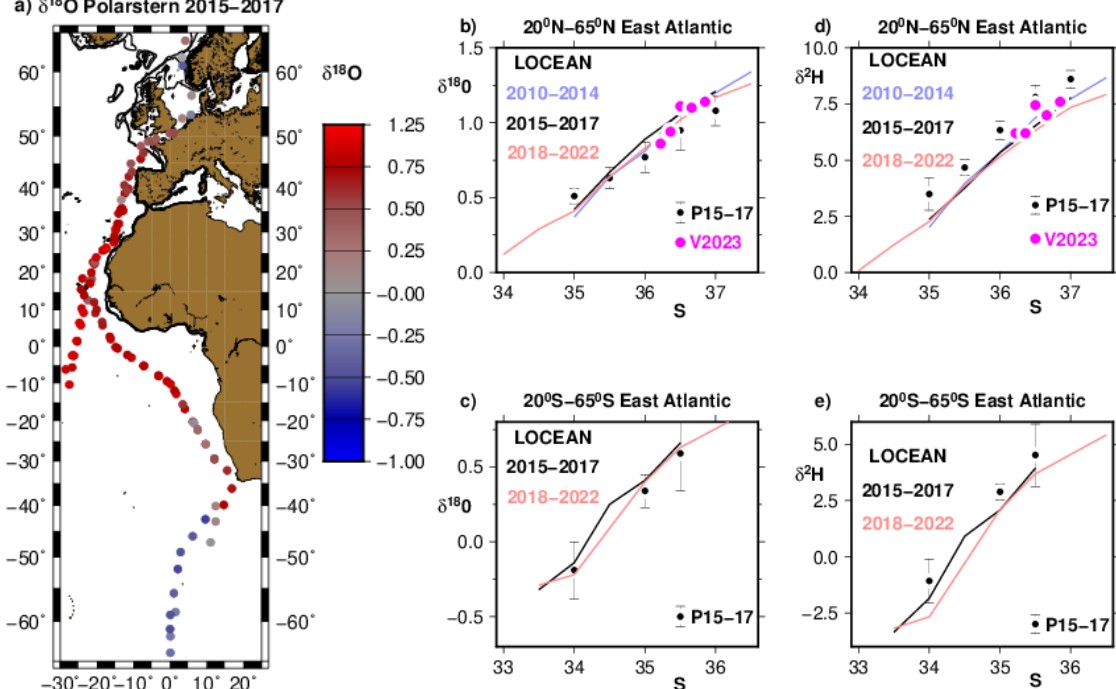


Figure 1: Comparison of the LOCEAN and Bonne et al. (2019) datasets. (a) map of RV
Polarstern dataset points east of 30°W in the eastern Atlantic Ocean. (b), (c), (d), (e)
Water isotopes-S scatter diagrams averaged as a function of salinity in 0.5 practical
salinity bins ((b) and (c) for $\delta^{18}$O; (d) and (e) for $\delta^2$H), top for the northern hemisphere
and bottom for the southern hemisphere, east of 30°W and outside of [20°N, 20°S]. The
black dots are the binned averages of the Bonne et al. (2019) RV Polarstern data in
2015-2017 (after adjustment of +0.25‰ to $\delta^{18}$O) (P15-17), with the root mean square
of the variance reported as error bars. Five individual surface points from Voelker et al
(2023) (V2023) are also plotted (magenta dots). The colored lines represent average
relationships of water isotopes in the LOCEAN data base in the same regions as a
function of practical salinity for three different period ranges.
The average relationships found in the LOCEAN dataset for three periods overlay well in
particular in the northern hemisphere. Uncertainties on individual curves (not shown)
are estimated based on the scatter of individual data in each salinity bin. They are
typically on the order of 0.01-0.02 (0.05-0.10) ‰ for $\delta^{18}$O ($\delta^2$H) respectively in the
northern hemisphere (top panel), and a little larger for the less sampled southern
hemisphere curves in 2015-2017. Sampling is usually also insufficient at the low end of
the salinity range, to reliably estimate an uncertainty. Thus, these different curves nearly
overlay within the sampling uncertainty. Five surface samples that were collected in the
Northeast Atlantic during the same years within the same salinity range (Voelker et al.,
2023), also fit well on the North Atlantic curves. The adjusted $\delta^{18}O$ data from Bonne et
al. (2019) are slightly shifted downward with respect to the curves (Fig. 1b, c), with the
plotted standard deviation of individual data around the average not overlapping the
LOCEAN data average curves in most cases for the same years 2015-2017. The situation
is opposite for the 35-salinity bin in the northern hemisphere, with the adjusted $\delta^{18}O$
data from Bonne et al. (2019) being above the three LOCEAN average curves, which
might be due to samples collected uniquely in the English Channel and North Sea by RV
Polarstern in this salinity range, whereas sampling is more geographically-spread in the
LOCEAN data base.
Altogether, the average $\delta^{18}O$ offset is small, with the LOCEAN data being higher by 0.02 ±
0.01 ‰ than the $\delta^{18}O$ from Bonne et al. (2019), which is not significantly different from
0 based on the interannual differences witnessed in the LOCEAN curves and the
scatter/uncertainty in the RV Polarstern data. A systematic difference is, however, found
for $\delta^{2}H$, with LOCEAN data been lower than $\delta^{2}H$ from Bonne et al. (2019) by 0.99 ±
0.07‰ (Fig. 1d, e).

2.2 Southern subtropical Indian Ocean
GWS2024 describe a synthesis of water isotope data in the southern Indian Ocean
combining their data collected in 2018 in the southeastern Indian Ocean (CROCCA-2S)
with earlier data in the southwestern Indian Ocean, in particular from LOCEAN, as well
as data from the southern Australian shelf collected mostly in 2010 (Richardson et al.,
2019), and in the equatorial Indian Ocean (Kim et al., 2021). In the most recent version
of the LOCEAN dataset, in addition to data included by GWS2024 and collected mostly
west of 80°E, there are two transects with surface data through the southeastern Indian
Ocean, one collected in February 2017, and the other in March 2024, thus in mid to late
austral summer. These transects cross the region covered by the CROCCA-2S dataset,
albeit not close to western Australia, as well as the area of the Richardson et al. (2019)
dataset, south of Australia. The LOCEAN dataset also contains surface data south of
Tasmania (in 2017, as well as in 2020 to 2024). All these data correspond to samples
analyzed on a CRDS Picarro L2130 at LOCEAN, and with the protocols discussed by
Reverdin et al. (2022). The bottles in which the samples were stored were the same ones
for most of the samples, and time between collection and analysis varied, but was mostly
on the order of 6 months or less. Thus, this is a homogeneously produced set of data for
the years 2016-2024, which spatially and temporally overlaps with the data used by
GWS2024 collected south of Australia and in the southeastern Indian Ocean (Fig. 2).

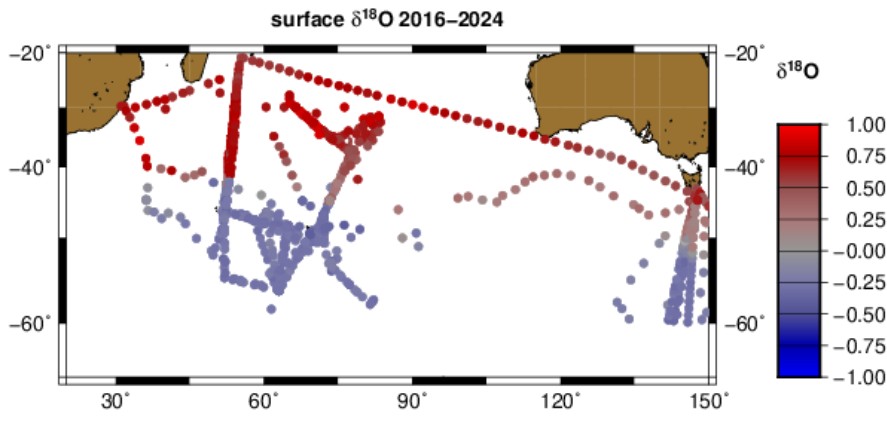



Figure 2: Map of $\delta^{18}O$ surface data in the LOCEAN archive for 2016-2024, north of 60°S.
All data are associated with S and $\delta^2H$ measurements.
The LOCEAN data distribution plotted in the S$\delta^{18}O$ space presents a wide $\delta^{18}O$ range at a
given salinity in the southwestern Indian Ocean (Fig. 3a) for S between 35 and 36. For
this range which covers a large part of the surface water of the southwestern Indian
Ocean's subtropical gyre, we establish a regression line for the LOCEAN $\delta^{18}O$ as a
function of S, which can be seen as a mixing line. Above this line, there are no data points
for lower S (Fig. 3a), with data at higher S found north of 28°S as well as in the far
southwestern Indian Ocean, but with some remnants found all the way to the core of the
subtropical gyre near 75°E/35°S (Fig. 3b). Data below the regression line contain most
of the data east of 60°E for latitudes south of 28°S and connect the salinity maximum
region with the lower salinity south of the Subtropical Front and down to the region
south of the Polar Front (Fig. 3c).  These subtropical lower isotopic values in S-$\delta^{18}O$
space, which already appear in part of the repeated (1998-2024) French OISO cruises
data at 50°E, dominate east of 60°E.

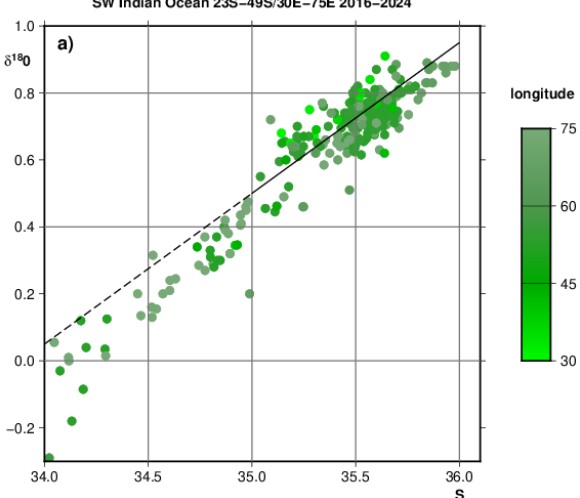


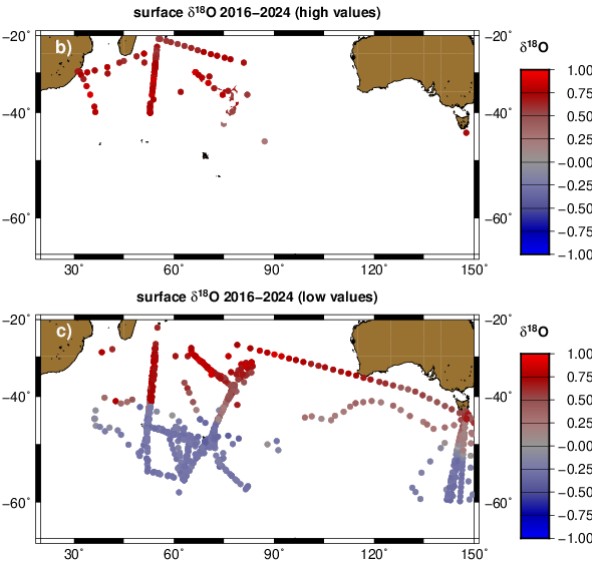


Figure 3: (a) S-$\delta^{18}$O scatter diagram of 0-30m LOCEAN data within the southwestern
region (30-75°E/23-49°S) coloured as a function of longitude, with the regression line
(black line) of the data in S-$\delta^{18}$O space for the 35-36 range in practical salinity. The
spatial distributions of the LOCEAN data with higher and lower $\delta^{18}$O relative to that
regression line in the whole Indian Ocean north of 60°S are shown on panels (b) and (c),
respectively.
We will now focus on the lower part of the distribution in S-$\delta^{18}$O space (Fig. 3c), which
overlaps with the location of the data from CROCCA-2S and the near-Australia data from
GWS2024 (the higher values in Fig. 3c do not). For salinities above 35 one observes a
lowering of δ¹⁸O at given salinity from 50°E in the western Indian Ocean to at least
100°E (Fig. 4) with more stable values further east. This lowering is on the order of 0.15
at most, even for the higher salinities (35.5 or more) for which it is strongest.

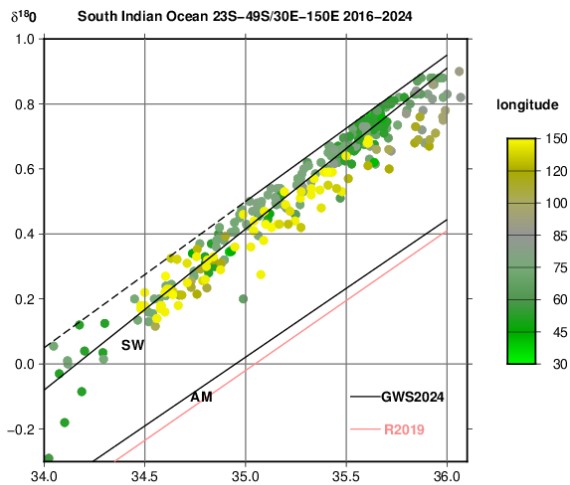


Figure 4: S-δ¹⁸O scatter plot of 0-30m LOCEAN Indian Ocean data as shown in Fig. 3c,
color-coded as a function of longitude, below the partially stippled regression line for
the SW Indian Ocean (reproduced from Fig. 3a). The two black lines correspond to the
two linear relationships (GWS2024) for the 0-100m layer between 23°S and 49°S for the
south-west Indian Ocean (SW) and for the Australian margin south of Australia (AM)
(we use the original relation of δ¹⁸O = 0.4231 * S - 14.7876, instead of the rounded-up
relation reported in the paper; R. H. Glaubke, pers. comm., 2024), and the pink line is the
earlier linear relationship for the 0-600m layer along the Australian margin by
Richardson et al. (2019) (R2019).
Thus, besides some gradual and smaller changes, we do not observe in the LOCEAN
surface dataset a large sudden change in the S-δ¹⁸O distribution near 75°E or 85°E
between the southeastern and southwestern Indian Ocean, nor a further strong change
closer to the Australian coastal margin, as suggested by figures 6 and 7 of GWS2024.
Most of the LOCEAN S-δ¹⁸O data south of 28°S correspond to the mixing of a low salinity
end-member characteristic of the fresh waters of the Southern Ocean (at S < 34) with
waters which are imprinted by air-sea exchange in the subtropical gyre at higher
salinities up to 36 and more, as discussed by GWS2024. These LOCEAN (S, δ¹⁸O) values
are significantly above the linear relationships proposed by GWS2024 (based on their
figures 5a, 6 and 7). This positive offset at given S seems to be about 0.05-0.10 ‰ in the
southwestern Indian Ocean, but close to 0.50 ‰ for the Australian coastal margins,
although we could not access the individual data of R2019 for that latter region. These
offsets are much larger than the spread in the LOCEAN data, which is on the order of
0.10 ‰. Furthermore, the LOCEAN data support the presence of a secondary low
salinity end member at S < 35 with heavier isotopic composition, contributing to the
water-mass properties in the far southwestern Indian Ocean as well as for the area
sampled between 20°S and 28°S north of the subtropical salinity maximum.  This could
be a contribution of the Indonesian Through Flow and tropical western Indian Ocean
surface waters, as discussed by Kim et al. (2021) and GWS2024. We could not carry out
a comparable comparison for $\delta^2H$ which is not presented by GWS2024, and which
exhibits a too large spread in the CROCCA-2S dataset to reach a firm conclusion.
3. Discussion
In the two intercomparisons of surface data presented in this note, we find significant
differences between datasets. Do these differences originate from spatio-temporal
variability or from systematic offsets between the different datasets?
In the case of the RV Polarstern dataset (Bonne et al., 2019), an error in a specified
reference material value was found after the publication, and the adjusted data present
only a small, non-significant $\delta^{18}O$ negative offset, but a significant positive $\delta^2H$ offset
with respect to LOCEAN data. Differences might arise from spatial differences. For
example, in the northern hemisphere, values at salinity close to 35 mostly originate from
the North Sea and English Channel in the RV Polarstern dataset, thus with more mid-
latitude continental influence than for most of the LOCEAN data in the same salinity
range which have a contribution of more depleted subpolar and polar freshwater. One
expects a larger isotopic range in the South Atlantic for salinities less than 35, due to
intermittent presence of sea ice or iceberg melt, and at higher salinities due to the
presence of different water masses originating from the South Atlantic and southeastern
Indian Ocean. However, the current dataset is not sufficient to estimate it.
Furthermore, different seasons were sampled in the two datasets. In the northeastern
Atlantic sector, Bonne et al. (2019) surface data east of 30°W were collected in April and
November north of 10°S and in November south of 10°S in the southeastern Atlantic.
These data do not suggest large seasonal differences in the Northeast Atlantic,
concurring with the LOCEAN S-$\delta^{18}$O data in the tropics to mid-latitudes (20 to 50°N),
which are tightly distributed along a mean S-$\delta^{18}$O relationship, and thus with low
seasonal variability of this relationship (Benetti et al., 2017a; Voelker et al., 2015).  The
LOCEAN data are not numerous enough in the southeastern Atlantic to further evaluate
whether the offset is constant throughout the dataset, or presents a component related
to geographical temporal or spatial variability.
To investigate the South Indian Ocean seawater isotopic composition, GWS2024
combined datasets that were processed in different laboratories. Potential offsets
between those could thus cause apparent spatial variability. In particular, GWS2024
outline large spatial contrasts in the S-$\delta^{18}$O relationship across the surface subtropical
Indian Ocean and southern Australia that are at least a factor two smaller in the recent
version of the LOCEAN dataset.
Seasonal or interannual variability might contribute to the differences shown on Fig. 3,
as the data in the southeastern Indian Ocean from GWS2024 were collected in
November-December, whereas the data in the LOCEAN database in this region are
mostly from February-March. However, at least south of Tasmania, where the LOCEAN
dataset also contains December data, it does not seem that the seasonal cycle causes
changes larger than 0.05 ‰ at the same salinity. A difference due to seasonality would
thus be barely identifiable in that case, noting the possible presence of interannual
variability and that the long-term accuracy in the analyses in some centers, such as AWI
Potsdam and LOCEAN, is 0.05 ‰. Richardson et al. (2019) also commented that south of
Australia there was little difference between a southern winter cruise and late summer
(March) data. Further west, near 55-70°E, earlier surface data in the OISO surveys, as
well as the vertical upper profiles of OISO station data also suggest a rather modest
seasonal variability on the order of 0.10 ‰. Changes could also arise from interannual
variability, but the range of interannual variability in the LOCEAN data base is smaller
than the difference between the GWS2024 curves for the southeastern Indian Ocean and
south of Australia and the corresponding LOCEAN data. Thus, a likely cause of the large
differences between the South Indian Ocean/Australia margin data combined in the
GWS2024 study is the existence of systematic offsets between the data produced by
different institutes.
4. Conclusions
What these two comparisons suggest is that offsets are present between different recent
published datasets, which exceed 0.10 ‰ in $\delta^{18}O$ and 0.50 ‰ in $\delta^2H$, thus larger than
the target long-term accuracy of analyses in individual isotopic laboratories. Moreover,
errors in reference material values are always possible and require post-analysis
intercomparisons, such as the one that led to the correction of the RV Polarstern dataset
(Bonne et al., 2019). Furthermore, one contribution to a systematic difference between
the LOCEAN dataset and data from other institutes is that the LOCEAN data are reported
in 'freshwater' concentration scale (Benetti et al., 2017b). The use of this concentration
scale corrects possible effects of salt in the water activity measured by IRMS with $CO_2$-
equilibration and the effect of salt accumulation during evaporation in laser
spectroscopy, which both can lead to fractionation, possibly of similar magnitude
(Walker et al., 2016). Different comparisons based on duplicates collected during cruises
suggest that this is a main cause of difference between LOCEAN data and other datasets
(LOCEAN $\delta^{18}O$ data being more positive). Poor conservation of the samples during
storage, analytical protocols, or uncertainties in the specified values of reference
material are other sources of differences between data produced in different institutes.
Different methods have been used for intercomparing and detecting systematic offsets
between different datasets. One common approach is to compare values obtained in
specific water masses, for which we expect little variability of the water isotopic
composition. This is often attempted, but data density is often limited, and the resulting
uncertainties are difficult to assess. Datasets with intermediate and deep data in the
Southern Ocean might be valuable to systematically test this approach, and model-based
reconstructions of isotopic composition of sea water could also be incorporated.

An alternative, in particular for the surface data, is to develop approaches based on the
systematic comparison of nearby data in space and time. In some ways, the assumption
behind this and what was done in the mapping by LeGrande and Schmidt (2006), that is
that the bulk of the variability is from large scale relationships of water isotopes and
salinity. This is also what has been done by crossover analyses in major geochemical
databases, such as GLODAP, with an attempt to adjust offsets for $\delta^{13}C$-DIC with a similar
low-density data distribution in the North Atlantic (Becker et al., 2016). The comparison
presented here (Fig. 1) of the S-water isotopes surface distribution in the North and
South Atlantic of the LOCEAN and the RV Polarstern (Bonne et al., 2019) datasets
suggests that this can be used to estimate offsets. Required improvements, in particular
for estimating uncertainties would be to take into account estimates of seasonal,
interannual and spatial variability in these relationships. However, this requires that
there are enough overlapping data within regions of relatively homogeneous water
masses, or some independent estimates on these signals, for example from model
simulations.

As the spatial and temporal data density is often reduced, we expect that the
uncertainties in estimated offsets will be large. This could reduce the usefulness of the
isotopic data for different oceanographic and climate studies, with large uncertainties in
estimated S-$\delta^{18}$O (or S-$\delta^2$H) relationships to validate proxies used for paleo-climate
reconstructions, or for identifying emerging climate-change related signals.

Scientific Committee of Oceanic Research (SCOR) working group 171 MASIS (Towards
best practices for Measuring and Archiving Stable Isotopes in Seawater) has recently
been established to contribute tackling these issues, both for water isotopes and the
isotopic composition of inorganic carbon in sea water, $\delta^{13}$C-DIC. For that, it aims to
actively involve the international community in establishing guidelines for data
production (collection, storage, measurement) and quality control, as well as for
validating the data and comparing well-documented archived data originating from
different laboratories. It will review the methods to estimate errors and offsets between
the different datasets. An important step for this effort is to directly intercompare
measurements by the different laboratories of shared well-preserved water samples
distributed quickly, as had earlier been done for $\delta^{13}$C-DIC (Cheng et al., 2019). This,
together with enhancing interaction within the scientific community needs to be actively
pursued, in order to reduce the errors when merging different datasets and increase the
potential use of the water isotope data.

Data availability
The LOCEAN data are available at https://www.seanoe.org/data/00600/71186/.
The isotopic data of the Bonne et al. (2019) are available as indicated in the paper, with
here S added from the PANGAEA archive, as described in the text. The GWS2024 data
are available as described in the paper. However, among the data used in this paper, we
could not access the data from the Richardson et al. (2019) paper.

Author contribution: GR initiated the study and prepared the manuscript with
contributions from all coauthors. AV initiated the intercomparison effort, and AV, CW,
and HM contributed to editing the paper. HM was also responsible from producing the
data in the Bonne et al. (2019) paper.

Competing interests: The authors declare that they have no conflict of interest.

Acknowledgments
The LOCEAN isotopic laboratory is supported by OSU Ecce Terra of Sorbonne Université.
We are thankful to Catherine Pierre and Jérôme Demange who have set and help run the
facility, and for Aïcha Naamar, Marion Benetti and Camille Akhoudas to have measured
some of the water samples. We are grateful for support by INSU, Nicolas Metzl and Claire
Lo Monaco for samples during the OISO cruises on RV MD2, by IPEV during the SOCISSE
program on RV Astrolabe, with on board support by Patrice Bretel and Rémi Foletto, and
by IPSL for supporting the LOCEAN data base and intercomparisons. Antje Voelker
thanks Joanna Waniek (IOW, Germany) for collecting the NE Atlantic water samples and
Robert van Geldern (GeoZentrum Nordbayern, Germany) for analyzing them. AV also
acknowledges financial support by Fundação para a Ciência e a Tecnologia (FCT) through
projects Centro de Ciências do Mar do Algarve (CCMAR) basic funding
UIDB/04326/2020 (https://doi.org/10.54499/UIDB/04326/2020) and programmatic
funding UIDP/04326/2020 (https://doi.org/10.54499/UIDP/04326/2020) and the
CIMAR associated laboratory funding LA/P/0101/2020
(https://doi.org/10.54499/LA/P/0101/2020). The RV Polarstern dataset was funded
by the AWI Strategy Fund Project ISOARC. Comments by Alexander Haumann (AWI) and
by two anonymous reviewers were very helpful.

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
