# Peer review of "Technical note: Large offsets between different datasets of seawater isotopic composition"

_EGUsphere, 2024_

## Author Comment (AC1)

Initial paper submitted to OS 25/09 2024

RC1

General comments: This short note presents some seawater oxygen and hydrogen isotope data
generated in the lead author's laboratory, and conducts a comparison with other datasets from
the same ocean regions generated in other labs. The major finding appears to be that there are
troubling differences in the data, possibly instrumental or procedural in origin. Whilst
concerning, this is not an especially surprising finding – I think that most scientists who work
with such data are quite attuned to the possibility of inter-laboratory offsets, and in many
cases will conduct their own ad-hoc comparisons to try and make datasets consistent where
possible.

Different instrumentation and protocols could be partly to blame, in some cases, and the paper
outlines the possible origins of some of these differences. I suspect that long-term
maintenance of standards is also an issue – while all samples are supposed to be measured
relative to known standards (e.g. VSMOW), the cost and availability of VSMOW is such that
samples are virtually never measured against it directly, but instead against intermediate
(secondary) laboratory standards, which are themselves measured relative to e.g. VSMOW.
Any drift or inaccuracy in the known composition of these secondary standards will thus feed
through into the data.

**Au:** we fully agree about the issue of drifting or uncertain secondary standards, or opening
too many times the small VSMOW standard bottle (we had mentioned 'uncertainties in
reference materials used', but will make that clearer). This could result in biases in estimated
isotopic values (often too low values). There are published recommendations (from IAEA) on
how to keep the secondary standards for a long time free of noticeable drifts, which some
groups (but not all) follow. Even though, we are aware of the importance to check at relative
intervals whether the standards are kept in good condition, and the possibility of small errors
due to that still sneaking in (in $\delta^{18}O$, possibly on the order of 0.01 to 0.02 ‰).

The current paper does a reasonable job of highlighting these sorts of issues in the context of
the datasets examined, but overall the treatment is relatively superficial. The analysis
essentially compares a couple of datasets and considers whether differences might be "real"
(i.e. deriving from oceanic temporal or spatial variability), before concluding that they are
most likely not. One could argue that an even more useful analysis would assess all available
public isotope datasets and conduct some sort of crossover analysis that would tabulate
offsets. (I believe such an activity is being conducted at AWI Bremerhaven, and I note the
acknowledgement to one of the key researchers there – perhaps there is scope to ramp up that
dialogue and deepen the analysis presented here, especially given AWI representation
amongst the authorship team for this paper?). This would enable the full extent of the problem
to be at least quantified, and possibly its root causes to be better identified, if the derived
offsets were categorised by method, protocol, instrumentation etc. I realise this is a much
bigger job than the authors intended to undertake, but I feel it would be significantly more
useful.

Au: We thank the reviewer for the feedback and insight. We are aware of two such efforts
(one at AWI (Bremerhaven, Ge), and one at LOCEAN-LSCE (Paris, Saclay, Fr)) to
systematically compare a large ensemble of different (water column) water isotopes data sets, and to characterize relative biases. This is a very valuable effort that we did not want to
duplicate in this study. We would like to point out that there is a large amount of data that are
surface (or near-surface/upper ocean only), in particular in tropical oceans (but also subpolar
gyres, and even in the Southern Ocean), and that are not included in these other investigations
(based on the surveys we have, probably more than half the data). The interest of our present
study, as we fully agree, just aimed at illustrating the issue on a few cases. Consequently, we
examine a few different data sets containing data rather close in time and space, albeit not as
close as wished for, which allows to quite well identify the systematic differences. This is
done to raise awareness, which does not seem to be shared by all data producers, to the best of
our knowledge, and to promote further crossover investigation in the recently approved
SCOR WG 171 – MASIS, an effort similar to what you propose.

Concerning ways forward, I think a minimum requirement of the paper should be a clear
statement on how the issue should be addressed in future. The paper alludes to some possible
methods that might be used to alleviate/address the problem (e.g. exchange of samples
between labs), but what is needed are firm recommendations and suggestions for who can
follow them up and (critically) who should oversee the process. For example, it is worth
noting that GO-SHIP has established protocols to deal with exactly this issue for other
variables to ensure their intercomparability, and might be well-placed to transfer/apply those
protocols to seawater isotope data also. Alternatively, possibly IAPSO has a role?

**Au**: we fully agree with that. This is one of the objectives of  SCOR WG 171 – MASIS
(Towards best practices for Measuring and Archiving Stable Isotopes in Seawater) that two of
the authors co-chair, and which is just being established this month. We have already set-up
different data exchange protocols, and started exchanging samples (and will include the AWI-
Bremerhaven austral ocean effort, piloted by Alexander Haumann, who is also a member of
the working group). As this is now official, we have added a paragraph on it to the note.

Comment (not included in the note): We have also been in touch with GO-SHIP co-
chairpersons, and will make a more official presentation on what we plan (the aim is to move
'water isotopes' from a parameter of class 3 to a parameter of class 2 in GO-SHIP). We are
also in touch with IAPSO (but for $\delta^{13}$C-DIC), and will liaise with the IOC UNESCO Ocean
Best Practices Group. We contemplated submitting last year a proposal on 'water isotopes' to
IAPSO, but timing was short, and we thought that it would weaken the SCOR WG proposal
and delay actions in too many different entities.

What is definitely not wanted is each lab or user conducting their own
intercomparison/correction exercises, since the resulting datasets (while internally more
consistent than before) will still not be comparable across them all, if different methods to
intercompare/correct are used.

**Au**: We fully agree. This was one of the main reasons why we decided to set up the SCOR
working group MASIS.

Overall, the paper highlights an issue that is concerning but not surprising. I have no objection
to the paper being published – I believe what it says is true, and the topic is an important one
– but a fuller treatment of the issue would be even more beneficial to the community,
including very specific recommendations on how it should be addressed.

**Au**: Thank you for your comment.

Minor points.

Title: perhaps should mention "… and suggested ways forward", or suchlike? Highlighting
the issue is important, but even more useful would be outlining what needs to be done to
resolve it.

**Au:** This is an interesting suggestion, but we feel that this is outlined in some way in the
second part of the title, and thus plan to keep the title as is.

Abstract: Is written from the context of comparing LOCEAN datasets to others, which is
sensible (I'm sure it is what the authors' starting point was) – but perhaps just saying
"intercomparing available public datasets" would be more balanced?

**Au:** We agree and changed the text accordingly.

Line 22. "carried out"?

**Au:** Done

Line 24. Punctuation is important here: "… between data sets, which exceed 0.1 in d18O and
0.5 in d2H, …" – the commas matter!

**Au**: Done

Page 1 and 2. Just a stylistic thing, but these paragraphs are really long… it would help the
reader to break them into smaller chunks.

**Au**: Thank you. This is implemented.

Line 123. When examining offsets, it's a bit unsatisfying that the isotope data from Polarstern
were not collected from the same waterline as the TSG. Some quantification of the impact of
this would be useful, especially if the sensors/intake were at different depths and/or different
positions on the hull.

**Au:** Yes, this is unfortunate, but the isotopic samples were probably collected there, because
of easier access to the water line. Author GR has been involved in studies of vertical near
surface ocean salinity stratification in a working group set up by spatial agencies (EA and
NASA). In the RV Polarstern data set, we found a few instances near Svalbard (Arctic Ocean)
and in the Southern Ocean where there was obvious salinity and water mass origin
stratification. This is also one reason why we restricted the comparisons to regions where
such large stratification is usually not taking place. Nonetheless, a salinity stratification of (at
most) a few 0.01 pss is always possible between the two levels, even in the best
circumstances. Such instances (mostly in the tropics) were commented in Jacqueline Boutin,
Yi Chao, William E. Asher, Thierry Delcroix, D. Drucker, et al., Satellite and In Situ Salinity:
Understanding Near-Surface Stratification and Subfootprint Variability. Bulletin of the
American Meteorological Society, 2016, 97 (8), pp.1391-1407. 10.1175/BAMS-D-15-

00032.1. hal-01360859 (and references here-in, in particular, Henocq et al. (2010): Vertical
variability of near-surface salinity in the tropics: Consequences for L-band radiometer
calibration and validation. *J. Atmos. Oceanic Technol.*, **27**, 192–209,
doi:10.1175/2009JTECHO670.1. (in that paper, one of the authors, GR, had compared the
salinities from the Polarstern two water lines, which were often available at the time).

Line 202. ACC fronts are usually capitalised – "Polar Front" etc.

**Au**: Thank you. Done.

Various places. "pss" seems to have crept in as a salinity unit. If the data are indeed measured
and presented on the practical salinity scale (as stated), then the salinities are ratios and hence
do not have units.

**Au:** We explicitly mention that practical salinity has no unit. We removed other later
mentions in the draft, but left it on the figures, as to not leave any doubt that what we plot is
practical salinity, according to the 1978 'scale'.

• **Citation**: https://doi.org/10.5194/egusphere-2024-3009-RC1
• **RC2**: 'Comment on egusphere-2024-3009', Anonymous Referee #2, 05 Nov 2024 reply

Review

**Technical note: Large offsets between different datasets of sea water isotopic**
**composition: an illustration of the need to reinforce intercalibration efforts**

Gilles Reverdin, Claire Waelbroeck, Antje H. L. Voelker, Hanno Meyer

This technical note highlights the important consideration of systematic offsets between
seawater isotopic values measured using different instrumentation and/or in different
laboratories. Isotopic measurements have been largely underutilized to-date, and being able to
reliably compare data collected and/or analyzed by different parties will be key in developing
a cohesive understanding of the ocean isotopic system.

The authors highlight the need for establishing "well-accepted systematic guidelines for

155 data production and quality control". Further, they advocate "enhancing scientific exchange
156 between the different institutes needs to be actively pursued, in order to reduce the errors
157 when merging different datasets". I strongly agree with these main
158 conclusions/recommendations, and feel that ongoing, wide-spread cross-calibration between
159 institutes and research groups is the only way to achieve this.

160 **Au:** Thank you.

162 While I am in agreement with the overall intention of the paper, I think it is difficult to make
163 this point, as presented, using surface samples alone. Some deeper digging beyond the offsets
164 being 'rather systematic' would help strengthen the case.

165 **Au:** (also, in response to comment by RC1) We agree that there is a strong interest in digging
166 further in particular with water column (and deep ocean) data. We are aware of two such
167 efforts (one at AWI and one at LOCEAN-LSCE) to systematically compare a large ensemble
168 of different (water column) water isotopes data sets, and characterize relative biases. This is a
169 very valuable effort that we did not want to duplicate in this study. We would like to point out
170 that there is a large amount of data that are surface (or near-surface/upper ocean only), in
171 particular in the tropical oceans (but also subpolar gyres, and even Southern Ocean), and that
172 are not included in these other investigations (based on what we are aware of, probably more
173 than half the data). The interest of our present study, which is only aimed at illustrating the
174 issue on a few cases, is that we have rather close in time and space data of the different data
175 sets, albeit not as close as wished for, which allows to quite well identify the systematic
176 differences. This is done to raise awareness, which does not seem to be shared by all data
177 producers to the best of our knowledge, and promote further crossover investigation, such as
178 the one you propose.

180 There are a few main points in the text that I feel could be addressed more carefully and/or
181 given some more thought and discussion.

183 **Main point 1**

184 This technical note focuses on surface water samples. Surface waters are much more variable
185 seasonally and geographically than deep water masses due to
186 evaporation/precipitation/freezing/melting. While this is acknowledged within the paper, I'm
187 not totally convinced that the offsets observed between the relatively limited datasets are
188 lab/method based rather than seasonal and/or geographic differences.

189 Au: Cf response above. We are fully aware of issues of the imprint from air sea (and sea-
190 ice/liquid water) exchange, which is what motivates some scientists (including some of the
191 co-authors) to have collected these surface samples. Based on our experience (one of the
192 authors, GR, has published different studies on evaporation isotopic properties, as well as
193 investigation on rainfall or sea ice melting/freezing imprint on near-surface isotopic
194 composition), this is what motivates the restriction of the domains upon which we intercompare the different data sets (tropics, sub-tropics and mid-latitudes to temperate
Southern Ocean areas). In these regions, and although one has to be quite careful, we have
some idea of what maximum seasonal cycle or interannual variability are in water isotope-
SSS coordinates averaged over regional scales. At first glance, they appear to be smaller than
when considering the surface variability at a fixed location, as salinity and isotopic
composition tend to co-vary seasonally. We are aware that we do not have the in-situ data to
fully test the impact of seasonal and interannual variability in the south-eastern Indian Ocean
or in the southern Atlantic Ocean in isotope-S space, but our guess is that this source of
variability is not the largest cause of the differences that we identify. This is an interesting
topic for further research in simulations of earth system models enabling water isotopes, but it
has not yet been done systematically, as far as we can tell (for example, simulations exist both
for MPI model (Xu et al., Geosci. Model Dev., 5, 809–818, 2012 www.geosci-model-
dev.net/5/809/2012/, doi:10.5194/gmd-5-809-2012), in iCESM (Brady et al., 2019, Journal of
Advances in Modeling Earth Systems, 11, 2547–2566.
https://doi.org/10.1029/2019MS001663), and in the IPSL model).

Without many similar datasets demonstrating the extent of natural variability, or direct
replicate analyses performed at different labs, it's difficult to make a convincing case that the
reported differences are analytical offsets and not observed natural geographic/temporal
variation. In absence of direct replicate analysis/cross-calibrations, the exercise detailed in
this paper may be better performed with deep water samples, with less variable isotopic
compositions.

**Au:** We fully agree that deep water samples are best for that, but we unfortunately do not
have similarly located deep water samples in the three data sets compared here. Deep samples
collected at the same site are rarely available, which is why other paradigms/approaches are
needed, as is been done in the Southern Ocean by at least two research groups. Although we
have only limited understanding of the observed natural geographic/temporal variation, we
found that the surface comparison approach is a complementary useful approach.

**Main point 2**

I have a hard time recommending the application of a 'correction' offset between datasets
without a direct cross-calibration between the two labs, it is impossible to know if the
difference in values is an offset (from technique, reference material, or sample evaporation),
or a true difference.

While correcting for a calibration offset between labs could be acceptable with appropriate
inter-lab cross-calibration efforts, trying to 'correct' data where samples may have been
compromised involves considerable risk, and could instill a false sense of confidence in
intercomparison efforts.

Au: (also cf response to comment RC1) We fully agree that direct 'cross-calibration efforts'
are required, and that is what we promote in the SCOR WG 171 – MASIS, which is just been
set up this month, and which is co-chaired by two of the co-authors of the paper.  Indeed, it is
in some instances difficult to fully identify the reason of the offset. However, in the LOCEAN
data set, there were some clear instances, in particular for earlier data sets but even during one
recent cruise, of sample evaporation having taken place; in these instances, if evaporation was low, an adjustment is proposed for cases when the suggested adjustment is less than 0.1/0.2 in
$\delta^{18}O/\delta^2H$ (with a flag specifying "probably good"). Furthermore, in the case of the RV
Polarstern data set analyzed at AWI-Potsdam, an offset in a reference material had been
identified, post initial data publication, which has been corrected before making the
comparison provided here. In other instances, reference material offsets happen that need to
be carefully identified to provide full confidence in the inter-comparison effort. We are,
however, aware that remaining offsets on the order of 0.01 to 0.02‰ in $\delta^{18}O$ due to
uncertainties in secondary reference materials might sneak in the data, even in the best
managed laboratories (at least, this is the case at LOCEAN).

More recently published material indicates that differences between analytical techniques (i.e.
IRMS vs CRDS) are insignificant (i.e. less than analytical precision). Reference material
errors can occur, and the only way to identify that for certain is cross-calibration between the
facilities in question. Unfortunately, facilities are often reluctant to spend the time/resources
on cross calibration, claiming that if all labs are (e.g.) referenced against VSMOW, then there
is no need (which is of course true, in theory, but not all labs operate the same way with
regards to calibration, replication, etc.). This is a problem that must be solved with buy-in
across the community, and a commitment to a longer-term vision of isotopic data (vs short-
term focus on a study from a single cruise, where offsets between labs/instruments are often
immaterial).

**Au**: We fully agree that this is an issue that must be solved with a long-term vision across the
community, and strongly support this buy-in vision. We'd like to point out that although there
are published indications that differences between analytical techniques (for $\delta^{18}O$ between
IRMS with $CO_2$ equilibration and CRDS (at least Picarro-based)) are less than analytical
uncertainties, this is not a universally recognized vision (L. Wassenaar, pers. communication,
or different papers on effects on the water isotope measurements of dissolved salts, including
one at LOCEAN by Benetti et al. (2017)). Thus, it might be by chance that they are so similar
(we believe that there are larger differences in the case of $\delta^2H$, or for other IRMS
measurement methods for $\delta^{18}O$, but this would need to be investigated further).

An offset resulting from sample compromise prior to analysis is the most difficult case.
Unless the offsets are very large, it is not truly possible to know which samples may have
been subject to evaporation during storage, or to what extent. Even if we can be confident that
some samples from a cruise were compromised, it cannot be assumed that each sample was
subject to the same amount of evaporation/fractionation offset. An attempt to correct some
number of collected samples from a dataset could have the unintended consequence of
unnecessarily offsetting samples within that set that were not actually subject to
compromise/evaporation. My opinion is that the best approach in this scenario is to discard
the data that has clearly been compromised (i.e. well outside of established natural variability)
and leave the rest alone.

**Au**: Cf response above. With CRDS measurements, we based this evaluation of sample
compromise (mainly, possible evaporation) on their d-excess value (which, if outside
acceptable ranges, is then flagged as probably bad), as well as on visual inspection of the
bottles and the caps. Users of the LOCEAN data base are free to discard samples that have been lightly compromised before analysis (based on the 'probably good' quality flag). Note
that we have chosen to retain them in the comparisons presented in this paper, as they have no
statistical effect on the comparisons provided.

Unless a direct inter-lab cross-calibration has been performed, I would rather not see
'correction' offsets applied to datasets in attempt to make them more comparable. Without the
cross-calibration data, there is simply no way to know whether one is truly making the
correction they think they are.

**Au**: Thank you for your opinion. We promote direct inter-lab comparison calibration, but
unfortunately this is not always available (this was not the case for one of the data sets of this
paper, despite our attempts), and furthermore, an a posteriori inter-comparison would not
necessarily be relevant to the earlier analyzed data set. Thus, we have to conclude that it will
be in some instances necessary to apply 'correction' offsets, as a secondary and more
uncertain approach to the more direct inter-lab comparison.

--

**A few specific minor points:**

**L40**: Glacial melt from ice shelves can impact isotopic composition well below the surface
ocean – down to 800m in the Amundsen Sea, Antarctica. (Biddle et al., 2019; Hennig et al.,
2024; Randall-Goodwin et al., 2015)

**Au**: Thank you; this was added.

**L66**: More recent studies have demonstrated significantly better precision with CRDS
systems - on par, or better than most published IRMS data. (0.025‰ in $\delta^{18}O$. Current
manufacturer-stated precision for Picarro systems is ±0.025‰ in $\delta^{18}O$ and ±0.1‰ for $\delta D$).
Voelker et al., (2015) achieved precision of ±0.06‰ $\delta^{18}O$ (in-run precision ±0.1‰ $\delta^{18}O$;
±1‰ $\delta D$), while Hennig et al., (2024) achieved precision of ±0.02‰ $\delta^{18}O$ (in-run precision
of ±0.04‰ $\delta^{18}O$). I'm not sure whether an advancement in technology, or a change in
methodology is responsible – but it doesn't seem that modern precision is meaningfully
different between IRMS and CRDS.

**Au**: Some groups in the past (still, now) provided IRMS data with similar 'high' precision,
but not all. I agree that there has been great progress with CRDS measurements, but there is
still sometimes a large issue of salt accumulation and its impact on memory effects during
runs. It is something that is present at LOCEAN, and for which the current solution is to run
reference materials through the runs with values close to the sample values analyzed. In that
respect, the situation is much worse for isotopic composition in salt water than in freshwater
(a solution that was earlier adopted at LOCEAN was to distillate the water samples prior to
analysis, but at great time cost). Note that memory effects could arise in IRMS measurements
too (but not due to salt).

**L78**: Is this significant? Other studies  (Hennig et al., 2024; Walker et al., 2016) showed equivalence within instrumental precision between IRMS and CRDS techniques. I'm not sure one can analytically justify applying an offset to data that is smaller than analytical precision.

**Au**: Yes, this effect can be highly significant, unfortunately (see comment above, based on experience gathered at LOCEAN on CRDS). Fortunately, this can be (partially) mitigated.

---

## Author Response (AR2)

Reviewer 2:

The authors have done a reasonable job responding to comments from the referees. The
focus of the paper has been significantly improved since the initial submission, however
there are some meaningful revisions to the presentation of and description of the results
that should be made before the paper can be published.

Please use a different colourbar for d18O than for Longitude, the use of only a single
colourbar throughout makes the figures much more difficult to interpret than they need
to be. Choose one colourbar for d18O, another for longitude, and use those throughout.
If possible, the same scale should also be used each time. As currently presented, using
same colourbar for every variable makes the figures much more difficult to interpret
than they should be.

In all cases, a continuous colourbar should be selected. The red-blue colormap creates
the artificial impression of a split between the data. Perhaps this is the author's
intention, with regards to isotope values above/below 0‰ vs VSMOW, however they do
not indicate any significance of 0‰ as a meaningful 'threshold' so a continuous
colourbar would still probably be more appropriate. In this case, I don't think it's
important that the scale runs from (e.g.) -1‰ to +1‰. If the range in d18O values being
presented is from (e.g.) -0.5‰ to +1.25‰, that would be a more helpful range to see
represented by the colourbar.

Au: As recommended, on figures we now use another color scale for the longitudes than
for the water isotope. We also slightly changed the 'water isotopes' color scale, with the
less intense near 0‰ values been now greyer and not too light. This might have
contributed to the reviewer's comment on the split in the distribution. The choice -1‰
to +1‰ was motivated indeed by the 0‰ value versus VSMOW being an expected near-
average ocean value. An additional reason for that choice wasto have a nearly similar
scale on figure 1a than on figures 2, and 3 (the only change for that scale is that it
extends to +1.25‰ which is nearly reached for some North Atlantic data). We agree that
the data distribution implies that there are few values below -0.5, but occasionally this
happens, and thus the scale extending to -1‰ allows us to plot such lower values.

The manuscript should be carefully copy-edited, as there are several inconsistencies throughout, e.g.:

-"pss" is still used several times throughout

Au: We removed the two remaining cases of pss. We agree that once we signify that we use practical salinity, we don't need to repeat it further.

-Language is inconsistent regarding directions (South-East, southeast, Southeast, south- west)

Au: Thank-you. We unified the notations with the use of southeast (or southwest)

throughout.

Specific comments:

L148: Please add a legend to Figure 1 defining the colored lines, back dots with error bars, and magenta dots. A legend will greatly improve the readability. The d18O scale on

Figure 1 is -1.0‰ to +1.25‰, while in all other cases the d18O scale is -1.0‰ to +1.0‰

Au: The only change between the color scale for different figures is that in Fig. 1a, it was extended from +1.00‰ (i.e. the upper limit on Fig. 2-3) to +1.25‰. The reason is that there are a few data in the North Atlantic over +1.00‰, which would not be plotted otherwise. We have added a legend in Fig. 1b-e plots to clarify what is presented:'LOCEAN' added over the indicated periods on the left side, and 'P15-17' and

'V2023' on the lower right corners for the Polarstern averages and the Voelker et al (2023), respectively. In the caption, we have replaced 'curves' by 'lines' and have added the panel letters.

L211: Please describe and make explicitly clear why you decided to draw a regression through points between 35 and 36 salinity, and why this line was used to split data. A

fairly strong linear relationship can be seen in Fig 3a down to salinity 34, and it's not clear why the regression was only drawn for salinities higher than 35, or why you exclude data from above the regression line in subsequent Figures. In L209 you mention

'scatter' above the line – and subsequently exclude data above that line. Has this data been excluded because of compromise, or is it a geographical exclusion drawn by salinity? Is it just to make Figure 4 look nicer? What is the reason for the regression through those points, or for focusing on only those data below the line?

Au: The reason we focused on the 35-36 practical salinity range for defining a mixing straight line in the southwest Indian Ocean is to select data in the subtropical gyre. It is within this range that Glaubke et al (2024) suggested that there were different water masses with fresher contributions originating from either further south or further north.

Taking the slope of this straight line in the limited 35-36 range avoids being overly influenced by the very large number of points in the LOCEAN dataset for the southern fresher surface waters. The distributions of LOCEAN data above and below this 'mixing'

line (which we extrapolate outside of the 35-36 range) end up not overlapping in the eastern and far western or northern parts of the Indian Ocean, while there is a large overlap in the southwestern Indian Ocean (this is the 'scatter' we were referring to), due to different surface water masses. This is why afterwards on figure 4 we only show the data points below this line which are the only ones of the LOCEAN dataset for the regional domain of the CROCCA-2S and Richardson et al (2019) data sets. We have rewritten this paragraph which obviously was not clear enough, based on the comments received.

L212: By nature of how the regression is drawn, it would be impossible for data falling above the line to have a lower salinity than 35, so this shouldn't be mentioned as a result. This could be described further while addressing the regression in above comment on L211.

Au: the estimated mixing line was extrapolated outside of the 35-36 domain (now plotted with a dashed line). Thus, the statement that there is no data above the line for S

< 35 is not a given and worth mentioning. Indeed, if data further north had been included in the data set, it would have probably included points with S lower than 35

above the mixing line.

L223: The use of the same colourbar for different variables on different scales makes

Figure 3 hard to read. Please select different colourbars for each variable.

Au: We agree and have changed the colorbar of Fig. 3.a with a yellow to green scale.

Similarly, we have slightly changed the colorbar for water isotopes (from blue to red through grey)

L230: The 'gradual lowering' would be much more clearly illustrated with a continuous colourbar. As currently presented, it's hard to see anything in those figures other than the stark north-south divide between positive and negative d18O values – it's very difficult to see the east-west trend that you're highlighting, when the red-blue divide is so much more prominent.

Au: The comment on the 'gradual lowering' was referring to figure 4, not figure 3 (where it is hard to identify it  as salinity also changes spatially). We expect that this lowering on Fig. 4 is now clearer with the yellow to green scale (we have also removed 'gradual' from the sentence).

L234: Please add a legend to Figure 4 describing each of the lines. It could also be helpful for context to plot the 35-36 salinity regression line on this figure.

Au: Thank-you. The lines are explained in the figure caption, and we added in the lower right corner of the plot a legend for the two types of lines.

Reviewer 1

The authors have addressed the technical points raised in the first round of reviews in an adequate way. I would say that they've done a fairly minimalist job of revision. The key substantive point from the first round of reviews, to my mind, was the question:

what is this paper for? The paper highlights a data intercomparability issue, which people who use such data are (I think) already quite aware of. I had thought the paper could serve as a "call to arms", to spur the community into taking the issue more seriously and addressing it via e.g. an IAPSO working group, or a GO-SHIP activity. But the revised paper and responses indicate that both of these are already underway (which is good to know). So, I'm left wondering even more what the purpose of the paper is. I don't mean to sound overly negative, I just struggle to understand what the raison d'etre for this paper really is.

Au: Thank-you. Our experience is that not every user of sea water isotopic data or of products derived from the data is aware of the issue. Although this is taken seriously by a large part of the data producers and some of the users, we selected these two examples to illustrate why it remains an important issue. The Glaubke et al (2024) paper is actually a case in hand for this being sidelined. The other comparison in the surface

Atlantic Ocean stemmed from our interest to merge the LOCEAN and Polarstern data for further investigations, which lead us to find out that there had been some internal standard issues and that more work was required before merging the two datasets.

Indeed, the original GISS *Global Seawater Oxygen-18 Database* is a wonderful assemblage of datasets and the mapping based on it by LeGrande and Schmidt (2006) is a valuable first guess in many world regions (albeit not in the deep Southern Ocean waters).

However, although there were already efforts to adjust some of the individual datasets combined in this data base, this is still rather inhomogeneous with offsets in some subsets of a similar nature to the ones described here.

That all said, I believe that what the paper says is true, and I agree that the general issue is an important one. So I don't think publication would cause damage, or mislead anyone, and I guess it would be one more small piece of evidence explaining why things like the IAPSO group are important, albeit retrospectively. But is that enough to warrant a publication in Ocean Science? I am not sure.

I don't feel I can do much more as a reviewer to help boost the likely impact of the paper.

The authors chose not to follow up my suggestion of conducting a broader study incorporating all publicly-available datasets, performing crossover analyses, tracking down likely causes of individual offsets etc - which I agree would be a much bigger job, but could be done as a contribution to the IAPSO working group etc. and would genuinely be what the community needs.

Au: We agree that what the reviewer proposes is a very valuable but much larger job that needs to be undertaken with the support of a wide community. We hope that the

SCOR WG and other ongoing efforts (such as for GO-SHIP) will contribute to that, and we added a summary in the conclusions on how this could take place.

I think it is now probably an Editorial decision to determine whether to accept or not – I

kind of feel that it's now a binary yes/no choice, with the direction of the decision depending on how useful to the community this paper is likely to be.

Editor

Your technical note was returned to both reviewers. Reviewer 1 remains sceptical about the value of the paper, whilst not disagreeing with any of the conclusions. Reviewer 2

finds the message of the paper worthwhile, but has some further recommendations to improve the clarity and impact of the paper (e.g. different colour scales for O18 and longitude); I would like you to consider these carefully.

I have read the paper myself as an O18 person myself. I share some of Reviewer 1's concerns that we knew much of this already, however I appreciate the nice comparisons that you have undertaken with the two published papers. In many ways this technical note provides a valuable commentary on those papers, and a cautionary note about taking any oceanographic data set at face value.

Au: thank-you very much.

I think you could further strengthen the paper by being more specific about what people can do now to improve the reliability of O18 data – the "call to arms" that Reviewer 1

mentions – whilst awaiting the deliberations of the SCOR working group. For example, are there any recommendations for collection or storage of samples that you could make? Or how regularly intermediate standards should be monitored or stored?

Although this technical note is clearly not intended to be a "best practices" paper (as might be an outcome of the working group), some forward looking suggestions might be helpful.

Au: (GR) I am personally very sensitive to the issues that you mention on collection, storage, and the monitoring of intermediate standards and storage, having myself stumbled on issues with them in the last thirty years.  (all Au) Based on the material already existing, we trust that the SCOR WG just been established will provide useful guidelines. It is our plan for the next three years to work on those, provide 'best practices' to be submitted to a wider community, and actively initiate intercomparison efforts. We have expanded on this in the conclusion section, as follows:

"The working group MASIS (Towards best practices for Measuring and Archiving Stable

Isotopes in Seawater) of the Scientific Committee of Oceanic Research (SCOR) has newly been established to contribute tackling these issues, both for water isotopes and the isotopic composition of inorganic carbon in sea water, $\delta^{13}$C-DIC. For that, it aims to actively involve the international community in establishing guidelines for data production (collection, storage, measurement) and quality control, as well as for validating the data and comparing well-documented archived data originating from different laboratories. It will review the methods to estimate errors and offsets between the different datasets. An important step for this effort is to directly intercompare measurements by the different laboratories of shared well-preserved water samples distributed quickly, as was done earlier for $\delta^{13}$C-DIC (Cheng et al., 2019). This, together with enhancing interaction within the scientific community needs to be actively pursued, in order to reduce the errors when merging different datasets and increase the potential use of the water isotope data."

However I recognise that the main message of this paper is not towards those generating data, but more towards those who download such data from data bases and assume them to be "correct". If you are able to strengthen this message in your revisions, that would be beneficial, and may go some way to allaying Reviewer 1's concerns.

Au: Thank you.

Some minor things:

Line 45 I would capitalise Southern Ocean as a name

Au: Thank-you. Done

Please remove pss in line 271

Au: Done

There are a few references referring to something "in" a reference; please replace these to "by" since references are to the authors (e.g. line 261)

AU: thank-you. Done.